# Diabetes Mellitus Mediates Risk of Depression in Danish Women with Polycystic Ovary Syndrome—A National Cohort Study

**DOI:** 10.3390/biomedicines10102396

**Published:** 2022-09-26

**Authors:** Dorte Glintborg, Tanja Gram Petersen, Katrine Hass Rubin, Marianne Skovsager Andersen

**Affiliations:** 1Department of Endocrinology, Odense University Hospital, DK-5000 Odense, Denmark; 2Institute of Clinical Research, University of Southern Denmark, DK-5000 Odense, Denmark; 3OPEN—Odense Patient Data Explorative Network, Odense University Hospital, DK-5000 Odense, Denmark; 4Research Unit OPEN, Department of Clinical Research, University of Southern Denmark, DK-5000 Odense, Denmark

**Keywords:** PCOS, depression, register based, Danish, antidepressants, ICD-10

## Abstract

Aim: To investigate the risk of depression in Danish women with PCOS compared to controls and possible mediators for depression in PCOS. National register-based study in Danish women with PCOS (PCOS Denmark, *N* = 25,203) and age-matched controls (*N* = 112,414). PCOS Odense University Hospital (PCOS OUH, *N* = 998) was a sub-cohort of women with PCOS with available clinical and biochemical results. The main study outcome was depression occurring after PCOS diagnosis. Depression was defined according to hospital ICD-10 diagnosis codes and/or inferred from filled medicine prescription of antidepressants. Diabetes, medical comorbidity, infertility, hormonal anti-contraception and low family income were entered as mediators in Cox regression analyses for depression. In PCOS OUH, PCOS characteristics (age, BMI, Ferriman-Gallwey score) were entered in Cox regression analyses. The median age at cohort entry was 28 (interquartile range (IQR) 23; 35) years. The median follow-up time to incident depression or censuring was 4.8 (IQR 2.2; 8.8) years in PCOS Denmark and 5.2 (IQR 2.4; 9.2) years in controls. Women with PCOS had a 40% increased risk of depression compared to controls (Hazard Ratio 1.42 (95% CI 1.38; 1.47). In regression analyses, diabetes, medical comorbidity, infertility, hormonal anticonception, and low family income were significant mediators of depression. Mediation analyses showed that the proportion of the association explained by diabetes was 12.5% (95% CI 10.4; 14.5). In PCOS OUH, BMI, waist and Ferriman-Gallwey score predicted development of depression. Conclusion: The risk of depression was increased in PCOS. Diabetes was an important mediator of depression in PCOS.

## 1. Introduction

PCOS is a common endocrine disorder in premenopausal women with an estimated prevalence of 10–15% [1]. The Rotterdam criteria define a PCOS diagnosis by hyperandrogenism (biochemical/clinical), anovulation and/or polycystic ovaries [2]. Exclusion of other causes is required [2].

Quality of life is impaired in women with PCOS [3,4,5]. The 36-Item Short Form Health Survey (SF-36) scores in women with PCOS were comparable to patients with diabetes mellitus and asthma [6,7]. In a meta-analysis [8], depressive symptom had a median prevalence of 36.6% (interquartile range (IQR) 22.3; 50.0%) in women with PCOS and the odds ratio (OR) for moderate to severe depressive symptoms compared to controls was 4.14 (95% confidence interval (CI): 2.68–6.52). Previous studies regarding symptoms of depression in PCOS were conducted in selected study cohorts [8], which could have affected study results. In a Danish national setting [9], the OR (95% CI) for a hospital diagnosis of depression was 2.8 (2.39; 3.34) in 19,199 women with PCOS vs. controls and the prevalence of depression diagnosis in PCOS was 1.4%. In a Swedish register-based study [10], the OR (95% CI) for diagnosis of depression was 1.56 (1.50–1.63) in 24,385 women with PCOS compared to controls. Diagnosis of depression and treatment with antidepressants could be associated with weight gain and PCOS-like phenotype [11]. Inclusion of women with diagnosis of depression before PCOS diagnosis in register-based studies [9,10] could lead to the overestimation of the OR for depression in PCOS. Therefore, studies on the risk of depression after PCOS diagnosis are needed. In one study, the hazard ratio (HR) for subsequent occurrence of depression diagnosis after PCOS diagnosis was 1.34 (1.29–1.40) in 26,251 South Korean women with PCOS compared to controls [12]. This finding remains to be reproduced in other study cohorts.

Antidepressants are first-line treatment in patients with moderate to severe symptoms of depression and medical treatment with antidepressants is often initiated and monitored by the patients’ general practitioner. Few patients with diagnosis of depression attend hospital clinics. Antidepressants were prescribed to 16.3% Danish women with PCOS [9,11], which corresponded to an OR of 2.0 (1.91; 2.11) for the prescription of antidepressants in women PCOS compared to age matched controls [9]. To our knowledge, there are no recent studies regarding the use of antidepressants in PCOS, and further data have been requested [13]. 

The mechanism for higher risk of depression in PCOS is discussed [5]. Overweight and abdominal obesity are present in approximately 75% women with PCOS [14], and central obesity is associated with low quality of life [15]. Hyper-inflammation and insulin resistance in PCOS are associated with higher risk of type 2 diabetes, infertility, and autoimmune diseases [16]. The Hazard ratio (95% CI) for having a diagnosis of type 2 diabetes in Danish women with PCOS was 4.0 (3.7; 4.3) and type 2 diabetes was diagnosed four years earlier in PCOS compared to controls [17]. The risk of depression is increased in individuals with diabetes and the diagnoses diabetes and depression are often co-existing [18]. Women with PCOS, who also have obesity and medical comorbidity, could represent a high risk group for depression, but this hypothesis needs to be tested. PCOS is often diagnosed as part of fertility treatment, and fertility treatment in itself could be associated with depression [19]. PCOS can be diagnosed in up to 80% women with hirsutism [14]. Hirsutism is associated with impaired quality of life in PCOS [15] and in the hyperandrogen phenotype of PCOS, the risk of depression could be increased [8]. Oral contraceptive pills regulate menstrual cycles, levels of SHBG increase and hirsutism improves [20]. More than 50% of Danish women with PCOS compared to 25% controls had prescription of oral contraceptives [9]. Hormonal anticonception is a risk factor for depression in the general population [21], but if hormonal anticonception mediates higher risk of depression in PCOS is undetermined [5]. 

Here, we studied depression diagnosis in women with PCOS compared to controls. The study design was novel as we included a national study cohort of women with PCOS and had access to medicine prescription data of antidepressants. Furthermore, possible mediators for depression could be investigated in the study cohort. 

## 2. Material and Methods 

The study was a register-based cohort study in two patient populations with PCOS and one control population (Figure 1). In Denmark, all citizens are provided a social security number (Central Person Register, CPR), which is issued at birth or at immigration to Denmark. The CPR links data from all Danish registers at individual level [22,23]. We used data from the civil registration system [23], the National Patient Register [22], the National Prescriptions Registry [24], and The Danish Income Statistics Register at Statistics Denmark [25]. **The civil registration system** maintains complete records of births, deaths and emigration status [23]. Information included date of emigration and death, if applicable. **The National Patient Register** contains data on all inpatient hospital contacts in Denmark after 1977 and outpatient hospital contacts after 1995 [26]. Since 2002, the National Patient Register contains contacts for private hospitals except private practice specialists and general practitioners. We extracted ICD-10 codes for primary and secondary diagnoses from the National Patient Register’s psychiatric and somatic parts. **The National Prescriptions Registry** includes all prescriptions filled at Danish pharmacies with complete record of Anatomic Therapeutic Chemical (ATC) code, date drug dispensed and number of drug packets [24]. Information on all prescriptions issued by prescribers, general practitioners or specialists were retrieved from 1995. **Socioeconomic registers at Statistics Denmark**: The Danish Income Statistics Register, established in 1974, includes income of anyone economically active in Denmark [25]. 

We have previously published data from the study cohort with follow up until 2012 [9,17,27,28]. The present dataset represents a new data draft with additional follow up until 2018. The two patient cohorts were PCOS Denmark and PCOS Odense University Hospital (OUH). **PCOS Denmark** was defined by women aged ≥12 years [29,30] with a diagnosis of hirsutism or PCOS according to the International Classification of Diseases (ICD)-10 (L680 and E282) from 1995 to 2018. The index year was defined as the first time a patient received the diagnosis L680 or E282. **PCOS OUH** was an embedded sub-cohort of PCOS Denmark. PCOS OUH consisted of premenopausal women evaluated at the outpatient clinic, Department of Endocrinology and Metabolism Odense University Hospital during 1997–2012 with the diagnosis codes PCOS (E282) or hirsutism (L680). Details regarding the study cohort has been published previously [9,17]. Routine evaluation included medical history, clinical examination, transvaginal ultrasound, and fasting blood samples [9]. 

**The control population** consisted of five randomly drawn women from the civil population register for each included woman in PCOS Denmark. Controls were matched according to age of their matched PCOS case and were assigned the index date (date of first PCOS diagnosis) of their PCOS case. Controls had to be alive and living in Denmark on the index date. 

**Exclusion criteria**: The PCOS diagnosis requires exclusion of diseases with symptoms similar to PCOS [2]. We excluded women with ICD-10 diagnoses E220 (acromegaly), E24 (Cushing’s syndrome), E25 (adrenogenital syndrome), and Q96 (Turner syndrome). Depression should occur after the index date and women with pre-existing depression up till 5 years before the index date were excluded from the PCOS and control cohorts. 

### 2.1. Definition of Study Parameters

#### 2.1.1. Outcome

The primary study outcome was depression diagnosed according to ICD-10 (A or B diagnosis) and/or use of antidepressants. ICD-10 codes for depression were F320-F339 (Major depressive disorder, mild, moderate, severe, single and recurrent episodes). Use of antidepressants was defined as at least one prescription and redemption of antidepressants: N06A. 

#### 2.1.2. Covariates

Age was calculated at the index date. Number of births was extracted from the National Patient Register using the diagnosis codes Z340 and Z348. 

**Covariates for mediation analyses:** We considered the following potential mediators: Diabetes, medical comorbidity, infertility, use of hormonal anticonceptives, and low socioeconomic status [5,8,31,32]. Covariates should be present before the outcome date. Medicine prescriptions should include one or more dispenses before the outcome date. 

**Diabetes** was defined as ICD-10 diagnosis code of diabetes E10: insulin dependent diabetes, E11: non-insulin dependent diabetes mellitus, E13: other diabetes, E14: unspecified diabetes mellitus, or O24: diabetes mellitus in pregnancy or prescription of antidiabetic drugs: A10 (antidiabetics) excluding prescription of metformin (A10BA02) [17]. 

**Medical comorbidity***:* The Charlson comorbidity index is based on 19 conditions [33,34] and presence of comorbidity was defined as Charlson comorbidity index ≥1. We excluded diabetes diagnosis from the Charlson comorbidity index as results for diabetes were presented separately. We applied hospital diagnoses according to the National Patient Register within five years before the index date according to the ICD-10 operationalization by Quan [34] using the updated weights from Quan et al. 2011 [35].

**Infertility** was defined according to ICD10 diagnosis codes: N97, Z35 and fertility treatment: G03GA (gonadotrophin) G03GB (ovulation stimulation). 

**Hormonal anticonception.** Oral contraceptives were defined by ATC codes G03AA and G03AB (combined estrogen and progesterone OCP) and G03HB01 (OCP containing cyproterone acetate). Gestagens included oral contraceptives (G03AC) and intrauterine device (G02BA03). 

**Family income** was based on family income in the index year, excluding tax and interests. We assessed family income according to tertiles (high, middle, low). 

### 2.2. PCOS OUH 

In PCOS OUH, clinical and biochemical data were available. We performed a supplementary analysis where we included BMI, Ferriman Gallwey score, total and calculated free testosterone, and fasting blood glucose at the index date as predictors for depression in regression analyses. Baseline data of PCOS OUH and applied assays have been presented previously [9,17]. Serum total testosterone and sex hormone binding globulin (SHBG) were analyzed using radioimmunoassay after ether extraction [36]. This method correlates closely with results from mass spectrometry. The intra-assay coefficient of variation (CV) for total testosterone was 8.2% and for SHBG 5.2%. Free testosterone levels were calculated from total testosterone and SHBG. Blood glucose was measured on capillary ear blood using HemoCue.

### 2.3. Statistical Analyses

Baseline characteristics were given as frequencies for categorical variables and continuous variables were presented as median (interquartile range). Baseline characteristics were compared by Wilcoxon Rank Sum test and chi-squared test. 

Incidence rates of depression were presented with 95% confidence intervals (95% CI) and the cumulative incidence function of depression was calculated. In survival analyses, the cohort was followed from index date until incident depression, emigration, death, or end of study period (31 December 2018), whatever came first. We handled PCOS status as a time-dependent variable and allowed controls to switch to cases during follow-up. The hazard ratios (HRs) of depression were estimated in crude and adjusted Cox regression analyses where 95% CIs were obtained by using clustered sandwich estimators. We checked that the non-proportional hazard assumption was not violated. All regression analyses included the matching factor age, and adjusted analyses included the mediators: diabetes, medical comorbidity, infertility, use of hormonal anticonception and family income. A counterfactual approach was applied where the HRs adjusted for potential confounders corresponded to the total effect of PCOS on the risk of depression, which could be decomposed into a controlled direct effect (CDE) and the proportion eliminated (PE) by eliminating the impact of individual mediating factors [37]. The CDE captured the influence of PCOS diagnosis on the risk of depression if the association between PCOS and mediator was hypothetically removed. In mediation analyses, we estimated CDE of potential confounders (diabetes, medical comorbidity, infertility, use of hormonal anticonception and low family income). PE was obtained by dividing the CDE by the total effect. The 95% CIs for CDE and PE were obtained by bootstrapping using 100 replicates. 

In PCOS OUH, we investigated baseline clinical and biochemical characteristics according to development of depression by comparing medians and performing Cox regression analyses. 

Stata 16 (StataCorp. 2019. *Stata Statistical Software: Release 16*. College Station, TX, USA: StataCorp LLC) was applied for data management and data analyses on remote VPN access to Statistics Denmark.

### 2.4. Ethics

According to Danish law, a register-based cohort study does not need approval from the local Ethics committee or Institutional Review Board. The Data Protection Agency and Statistics Denmark approved the study, project no 704175.

## 3. Results

Figure 1: Depression within five years before the index data was present in 16.3% women with PCOS (5137/30,340) and 10.7% controls (13,447/125,861). After exclusion of women with exclusion criteria and previous depression, the study cohort included 25,203 women with PCOS (PCOS Denmark) and 112,414 controls. 

**Study characteristics of the cohort until end of follow up (Table 1).** Women in PCOS Denmark and controls had mean age 28 years by study inclusion. The median follow up duration was 7.0 (IQR 3.2; 13.0) years in PCOS Denmark, 11.3 (IQR 6.8; 16.1) years in PCOS OUH and 7.5 (IQR 3.4; 13.9) years in controls. Women in PCOS Denmark vs. controls had higher prevalence of diagnosis codes of diabetes (10.5% vs. 3.5%), medical comorbidity (3.9% vs. 2.6%), and infertility (41.4% vs. 23.4%), all *p* < 0.001, and higher prevalence of nulliparity (50.3% vs. 49.5%), *p* < 0.022. Medicine prescriptions for antidiabetics (6.2% vs. 1.8%), fertility treatment (31.3 vs. 8.7%), and hormonal anticonception (80.6 vs. 78.2%) were more prevalent in women in PCOS Denmark vs. controls, all *p* < 0.001. Low family income was more prevalent in PCOS Denmark vs. controls (34.4 vs. 33.4%), *p* = 0.002. 

Women in PCOS OUH vs. women in PCOS Denmark had higher prevalence of diabetes and infertility and more prevalent use of antidiabetics, and hormonal anticonception (Table 1). Low family income was more prevalent in PCOS OUH vs. PCOS Denmark. 

**Development of depression during follow up (Table 2 and Figure 2):** The total incidence rate of depression was 26.37(25.68; 27.07)/1000 person years (PY) in PCOS Denmark, and 18.44(18.17; 18.70)/1000 PY in controls (*p* < 0.001). The cumulative incidence of depression in PCOS Denmark was 40.7/1000 PY and 31.1/1000 PY in controls (Figure 2). Diagnosis codes of depression occurred in 4.2% women in PCOS Denmark vs. 3.0% controls (*p* < 0.001) during follow up and use of antidepressants occurred in 21.8% women in PCOS Denmark vs. 16.2% controls (*p* < 0.001). The incidence rate of depression during follow up was 27.44 (24.48; 30.75) /1000 PY in PCOS OUH.

**Proportional Hazard regression analyses (Table 3):** The HR for development of depression was 1.42 (1.38; 1.47) in PCOS Denmark vs. controls, *p* <0.001. Mediation analyses showed that the proportion of the association explained by diabetes was 12.5% (95% CI 10.4; 14.5). The proportion of the association eliminated by medical comorbidity, infertility, use of hormonal anticonception, and low family income was modest and ranged from 0.4 to 2.3% (95% CI range 0.2; 4.3).

**Characteristics according to development of depression in PCOS Denmark and controls (Table 4):** Depression was diagnosed at similar age in PCOS Denmark vs. controls (median 29 years, *p* = 0.42). Within PCOS Denmark, women with development of depression had higher prevalence of diabetes (17.3 vs. 10.4%), comorbidity (4.9 vs. 3.6%), and low family income (47.9% vs. 30.5%) compared to women without development of depression, all *p* < 0.001. Women with development of depression (yes) in PCOS Denmark vs. controls had higher prevalence of infertility (46.0 vs. 28.9%), diabetes (17.3 vs. 6.4%), comorbidity (4.9 vs. 3.6%), and low family income (47.9 vs. 47.6%), all *p* < 0.001. 

**PCOS OUH and development of depression (Table 5 and Table 6):** At baseline, women in PCOS OUH had median BMI 26.8 (23.0; 32.2) kg/m^2^, and 60.4% had BMI ≥ 25 kg/m^2^ (Table 5). Women in PCOS OUH who developed depression had higher baseline BMI (27.7 vs. 26.4 kg/m^2^) and higher FG score (12 vs. 10) compared to women in PCOS OUH without depression, *p* < 0.05. In regression analyses, BMI (adjusted HR 1.02 (1.01; 1.04), *p* = 0.003), waist (adjusted HR 1.03 (1.00; 1.05), *p* = 0.041), FG-score (adjusted HR 1.02 (1.00; 1.03), *p* = 0.041), and fasting BG (adjusted HR 1.18 (1.00; 1.39), *p* = 0.055) were independent predictors of depression in PCOS OUH with estimates close to one (Table 6). 

## 4. Discussion

In the present national register-based study, we found that women with PCOS had 40% increased risk of depression compared to controls and 12.5% of this increased risk could be attributed to diabetes. During follow up, the incidence rate of depression was 26.4/1000 PY in women with PCOS and 18.4/1,000 PY in controls (*p* < 0.001). These data outline the importance of depression in women with PCOS and support that the risk of depression could be especially increased in women with PCOS and diabetes. 

The present data expand results from our previous register-based study [9], where we reported HR 2.8 for ICD-10 hospital diagnosis of depression in Danish women with PCOS and HR 1.9 for prescription of antidepressants in PCOS [9]. The study addressed several outcomes of PCOS and follow up ended in year 2015 [9]. In the present study, depression was the primary study outcome and women with depression before PCOS diagnosis were excluded from the study cohort. Therefore, we excluded 16.3% women from the PCOS cohort and 10.6% controls, which could explain our finding of a more modest HR for depression compared to our previous data [9]. These findings underline that medical antidepressive treatment and/or depression-associated increased weight may be linked to PCOS-like symptoms and can result in diagnosis of PCOS. Furthermore, our findings show that a high percentage of Danish women are treated with antidepressants (35.5% women with PCOS and 25.3% controls). We observed a slight elevated risk of a depression diagnosis in PCOS compared to controls in line with some previous papers [10,11,12,38], whereas up to eight times elevated risk of high depression scores in PCOS were reported by others [8,15,39]. The present data were nationwide, whereas inclusion of selected cohorts of women with PCOS attending a hospital unit could explain the high risk of depressive symptoms in some previous studies. We defined depression outcome by ICD-10 diagnosis codes and/or use of antidepressive medication. Treatment with antidepressants is only indicated in case of moderate or severe depression, whereas non-medical treatment should be applied in patients with milder depressive symptoms [40]. Based on the present data, we cannot conclude whether some women with PCOS could have milder depression symptoms without need for medical treatment; however, available data support screening for depressive symptoms in PCOS. 

When diabetes was entered as mediator in regression analyses, the HR for depression was attenuated and the risk proportion eliminated by diabetes was 12.5%. Within women with PCOS, the prevalence of depression was 17.3% in women with diabetes compared to 10.4% in women without diabetes. These findings supported that diabetes may contribute to the increased risk of depression in women with PCOS. Our finding of diabetes as an individual mediator for depression in PCOS represents a novel study finding as previous studies were underpowered to investigate this hypothesis. Depressive symptoms were more persistent in persons with comorbid diabetes and depression [41] and combined diabetes and depression diagnosis could have more adverse consequences on health outcome than each diagnosis alone [41]. Within PCOS OUH, BMI, waist, and fasting blood glucose predicted development of depression, which remained significant in adjusted analyses. Our results are in accordance with studies reporting higher BMI and HOMA-IR in women with PCOS and concurrent depression compared to women without depressive symptoms [8,12]. Furthermore, the positive association between obesity, insulin resistance and depression is well described [42]. Our findings support that women with PCOS in combination with a diagnosis of diabetes need special attention regarding depression; hence, screening for depression may be especially relevant in women with PCOS and high metabolic risk. 

Within PCOS OUH, high FG-score was associated with higher risk for development of depression independent of age and BMI. Hirsutism is known to predict decreased quality of life in PCOS [4] and women with depressive symptoms had 53% higher odds of hirsutism compared to women without depressive symptoms (OR: 1.53; 95% CI: 1.10, 2.12; six studies) [8]. Furthermore, we previously reported that hyperandrogenism was associated with shorter time to first prescription of antidepressants in PCOS [11]. Women with PCOS according to NIH criteria had higher risk of depression than women with PCOS diagnosed according to Rotterdam criteria [10], which further highlight the importance of PCOS phenotype for risk of depression. Infertility, medical comorbidity, and low family income were statistically significant mediators of depression. The risk proportion eliminated by these factors was, however, relatively limited, which indicate that the proportion of the association between PCOS and depression mainly are explained by other factors. 

Hormonal anticonception was a significant mediator of depression in the present study, but the risk proportion eliminated by hormonal anticonception treatment was only 0.4% (0.2; 0.7). In women with PCOS, the prescription rate of hormonal anticonception was lower in women who developed depression during follow up compared to women without depression (79.4 vs. 80.9%). Our findings support that hormonal anticonception is a mediator of depression in the general population, but use of hormonal anticonception did not explain higher risk of depression in women with PCOS. Oral contraceptive pills are considered the most efficient treatment modality for hirsutism and menstrual cycles in PCOS. Given the positive association between hirsutism and depression, improved hirsutism during oral contraceptives would result in improved quality of life. The present study did not support this hypothesis. In accordance, symptoms of depression and quality of life were unchanged despite more regular menses and decreased hirsutism scores in clinical studies [43,44]. One randomized study reported that weight loss predicted well-being during oral contraceptives and/or intensive lifestyle intervention [45]. It is possible that higher metabolic risk and weight gain in PCOS during oral contraceptives [46] could have counteractive effects on risk of depression, but more studies are needed to test this hypothesis. 

Strengths and limitations can be observed in this study. The study design was an important strength with our use of well-validated register data in combination with available clinical and biochemical data in an embedded study cohort. This design allowed us to test hypotheses that could not be evaluated in the national cohort. The PCOS diagnosis was obtained by available ICD-10 codes at hospital contacts. Some women in the control group could have undiagnosed PCOS, which could underestimate the risk of depression in PCOS. Different definitions have been applied to define PCOS and the Rotterdam criteria in 2003 [2] allowed milder phenotypes as part of the PCOS definition. A relatively lean Nordic study population was included. Development of depression differ according to ethnicity [32] and prescription pattern for antidepressants could differ between countries. Therefore, our results need validation in study populations with higher baseline metabolic risk. The attention and questioning by a doctor may increase diagnosis rate of depression and result in increased prescription of antidepressants in study populations with several doctor contacts such as PCOS. Surveillance bias could result in higher diagnosis rate of depression and overestimation of the incidence rate for depression in women with PCOS. Surveillance bias could be partly overcome by selecting women with many hospital contacts for other diseases as controls. This would, however, carry a risk of selecting a control group of less healthy women. We chose to omit women with diagnosis of depression before PCOS diagnosis, however PCOS and depression are often co-occurring, and this design could be discussed. The PCOS OUH cohort represented a study cohort of women with PCOS referred for our hospital clinic for evaluation and treatment for PCOS. These women could differ from women with PCOS attending their private practitioner or a private gynecologist. Women with infertility complaints were referred to the fertility clinic, which could result in selection of a specific PCOS phenotype. 

**Conclusion,** the risk of depression was increased in women with PCOS compared to controls. Diabetes is an important risk factor for depression in women with PCOS. 

## Figures and Tables

**Figure 1 biomedicines-10-02396-f001:**
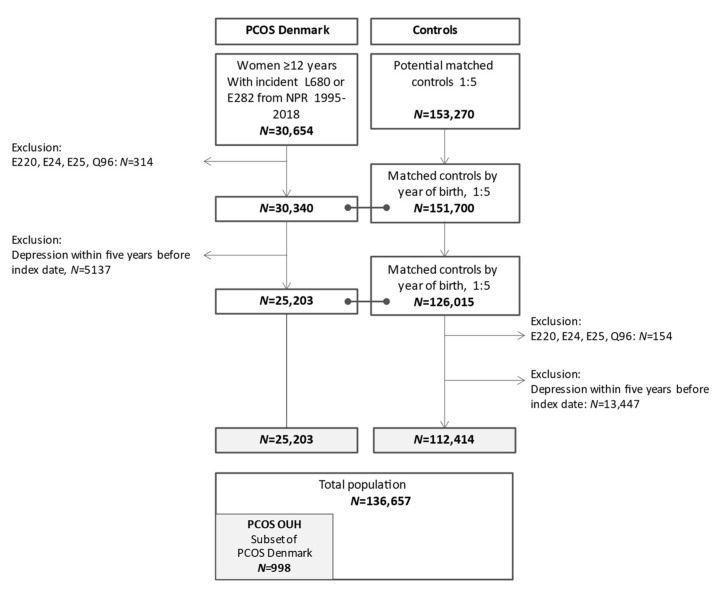
Flowchart of all included women with PCOS and controls.

**Figure 2 biomedicines-10-02396-f002:**
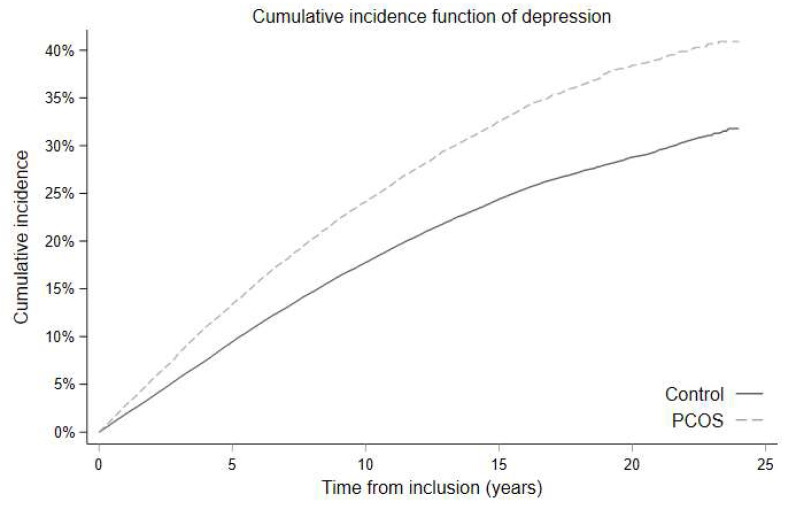
Cumulative incidence of depression in women with PCOS and controls in PCOS Denmark.

**Table 1 biomedicines-10-02396-t001:** Characteristics in women with PCOS and controls until end of follow-up (*N* = 136,657).

		PCOS OUH	PCOS Denmark	Controls		
		(*N* = 998)	(*N* = 25,203)	(*N* = 112,414)	*p* ^a^	*p* ^b^
Age (y) at diagnosis PCOS median (Q1–Q3)		28(21; 34)	28 (23; 35)	28 (22; 35)		
**ICD-10 codes**						
Diabetes mellitus	E10,11,13,14, O24	132 (13.2%)	2642 (10.5%)	3910 (3.5%)	0.004	0.000
Comorbidity at index		26 (2.6%)	979 (3.9%)	2944 (2.6%)	0.033	0.000
Infertility	N97, Z35	494 (49.5%)	10,443 (41.4%)	26,352 (23.4%)	0.000	0.000
Number of births	0	531 (53.2%)	12,680 (50.3%)	55,660 (49.5%)	0.062	0.022
	1	171 (17.1%)	4497 (17.8%)	16,850 (15.0%)	0.551	0.000
	≥2	296 (29.7%)	8026 (31.8%)	39,904 (35.5%)	0.130	0.000
**Medicine prescriptions**						
Antidiabetics	A10	78 (7.8%)	1552 (6.2%)	1996 (1.8%)	0.026	0.000
Fertility treatment	G03GA, G03GB	259 (26.0%)	7885 (31.3%)	9736 (8.7%)	0.000	0.000
Hormonal anticonception	G03AA, G03AB G03HB01, G03AC G02BA03	848 (85.0%)	20,313 (80.6%)	87,940 (78.2%)	0.000	0.000
**Combined covariates**						
Diabetes		143 (14.3%)	3015 (12.0%)	4399 (3.9%)	0.019	0.000
Infertility		525 (52.6%)	11,589 (46.0%)	28,603 (25.4%)	0.000	0.000
**Family income at index year**						
High		141 (14.1%)	8540 (33.9%)	37,170 (33.1%)	0.000	0.013
Middle		356 (35.7%)	7994 (31.7%)	37,706 (33.5%)	0.006	0.000
Low		501 (50.2%)	8669 (34.4%)	37,538 (33.4%)	0.000	0.002

Characteristics at the index date. Comorbidity defined as Charlson index ≥1. *p* ^a^ PCOS OUH vs. the rest of PCOS Denmark. *p* ^b^ PCOS Denmark vs. controls. Chi-square test (categorical variables) and non-parametric test on the equality of medians (continuous variables).

**Table 2 biomedicines-10-02396-t002:** Risk of development of depression in PCOS OUH, PCOS Denmark and controls after the index date.

	PCOS OUH (*N* = 998)		PCOS Denmark (*N* = 25,203)		Controls (*N* = 112,414)		
	*N* (%)	Incidence Rate per 1000 PY	*N* (%)	INCIDENCE Rate per 1000 PY	*N* (%)	Incidence Rate per 1000 PY	*p* ^
**ICD-10 depression**							
Depressive episode (F320-F329)	17 (1.7%)	1.12 (0.68; 1.86)	316 (1.3%)	1.22 (1.09; 1.36)	978 (0.9%)	0.84 (0.79; 0.90)	0.000
Major depressive disorder, recurrent (F330-F339)	54 (5.4%)	4.16 (3.18; 5.43)	742 (2.9%)	2.92 (2.71; 3.14)	2392 (2.1%)	2.08 (2.00; 2.17)	0.000
**Total ICD-10 (F32-F33)**	71 (7.1%)	5.35 (4.23; 6.78)	1058 (4.2%)	4.19 (3.94; 4.45)	3370 (3.0%)	2.94 (2.85; 3.05)	0.000
**Antidepressants**							
SSRI (N06AB)	224 (22.4%)	18.64 (16.29; 21.31)	4071 (16.2%)	18.18 (17.62; 18.75)	13,449 (12.0%)	12.81 (12.60; 13.03)	0.000
MAO inhibitors (N06AF, N06AG)	0 (0%)	NA	10 (0.0%)	0.04 (0.02; 0.07)	34 (0.0%)	0.03 (0.02; 0.04)	0.449
Other (N06AX)	160 (16.0%)	12.80 (10.95; 14.97)	2693 (10.7%)	11.26 (10.84; 11.70)	8542 (7.6%)	7.76 (7.59; 7.92)	0.000
TCA (N06AA)	83 (8.3%)	6.31 (5.08; 7.85)	1426 (5.7%)	5.73 (5.44; 6.04)	3942 (3.5%)	3.46 (3.35; 3.57)	0.000
**Total (N06A)**	299 (30.0%)	26.55 (23.65; 29.80)	5495 (21.8%)	25.81 (25.12; 26.51)	18,231 (16.2%)	18.02 (17.76; 18.28)	0.000
**Total event rate of depression**	307 (30.8%)	27.44 (24.48; 30.75)	5598 (22.2%)	26.37 (25.68; 27.07)	18,613 (16.6%)	18.44 (18.17; 18.70)	0.000

^ Chi square test between PCOS Denmark and controls.

**Table 3 biomedicines-10-02396-t003:** Hazard ratios for development of depression in PCOS Denmark (*N* = 18,476) and controls (*N* = 54,757).

Exposure	Mediating Factor	HR_Total_ (95% CI)	HR_CDE_ (95% CI)	HE_PE_ (95% CI)	PE (% (95%))
PCOS		1.42 (1.38; 1.47)			
		0.000			
	Diabetes mellitus		1.37 (1.33; 1.41)	1.04 (1.03; 1.04)	12.5 (10.4; 14.5)
			0.000		
	Medical comorbidity		1.42 (1.37; 1.46)	1.00 (1.00; 1.01)	1.5 (1.0; 2.0)
			0.000		
	Infertility		1.41 (1.37; 1.46)	1.01 (1.00; 1.01)	2.3 (0.3; 4.3)
			0.000		
	Hormonal anticonception		1.42 (1.38; 1.46)	1.00 (1.00; 1.00)	0.4 (0.2; 0.7)
			0.000		
	Low family income		1.42 (1.37; 1.46)	1.00 (1.00; 1.01)	1.5 (0.2; 2.8)
			0.000		
	All mediating factors		1.35 (1.31; 1.39)	1.05 (1.04; 1.06)	17.1 (14.3; 19.8)
			0.000		

Hazard ratios (HR) estimated using Cox regression, presented for the total effect (HR_Total,_ adjusted for age) and controlled direct effect (HR_CDE_, adjusted for age and mediating factor). Infertility, diabetes, medical morbidity (Charlson comorbidity index), and hormonal anticonception infertility are included as binary variables and family income according to tertiles. Hazard ratio for the proportion eliminated (HR_PE_) by the impact of mediator is calculated as HR_Total_/HR_CDE_. Proportion eliminated (PE) by the impact of mediator is only presented if the direction of HR_CDE_ and HE_PE_ is the same and calculated as (HR_Total_ − HR_CDE_)/(HR_Total_ − 1) ∗ 100.

**Table 4 biomedicines-10-02396-t004:** Characteristics in women with and without development of depression.

	Development of Depression in PCOS			Development of Depression in Controls					
	Total	Yes A	No B	Total	Yes C	No D	PCOS vs. Controls	P A vs. B	P A vs. C
**Development of depression**	25,203	5598	19,605	112,414	18,613	93,801			
Age at diagnosis (y) Median (Q1, Q3)	28 (23;35)	29 (23;36)	28 (23;35)	28 (22;35)	29 (23;36)	28 (22;34)	0.000	0.000	0.415
	N (%)	N (%)	N (%)	N (%)	N (%)	N (%)			
Age <20 years	3724 (14.8%)	779 (13.9%)	2945 (15.0%)	17,953 (16.0%)	2771 (14.9%)	15,182 (16.2%)	0.000	0.040	0.000
**Characteristics**									
Diabetes mellitus	3015 (12.0%)	967 (17.3%)	2048 (10.4%)	4399 (3.9%)	1189 (6.4%)	3210 (3.4%)	0.000	0.000	0.000
Comorbidity	979 (3.9%)	274 (4.9%)	705 (3.6%)	2944 (2.6%)	661 (3.6%)	2283 (2.4%)	0.000	0.000	0.000
Infertility	11,589 (46.0%)	2576 (46.0%)	9013 (46.0%)	28,603 (25.4%)	5388 (28.9%)	23,215 (24.7%)	0.000	0.954	0.000
Hormonal anticonception	20,313 (80.6%)	4443 (79.4%)	15,870 (80.9%)	87,940 (78.2%)	14,463 (77.7%)	73,477 (78.3%)	0.000	0.008	0.057
Low family income	8669 (34.4%)	2681 (47.9%)	5988 (30.5%)	37,538 (33.4%)	8856 (47.6%)	28,682 (30.6%)	0.000	0.000	0.000

Comorbidity was defined as a Charlson index ≥1. *p*-values obtained with chi-squared test for categorical variables and non-parametric test on the equality of medians for continuous variables.

**Table 5 biomedicines-10-02396-t005:** Baseline clinical and biochemical characteristics according to development of depression in PCOS OUH (*N* = 998).

	All		Depression		
			Yes *N* = 307	No *N* = 691	
Baseline Characteristics	*N* (%)	Median (Q1–Q3)	Median (Q1–Q3)	Median (Q1–Q3)	*p* #
Age (years)	998 (100%)	28 (21; 34)	29 (22; 35)	27 (21; 34)	0.058
BMI (kg/m^2^)	930 (93.2%)	26.8 (23.0; 32.2)	27.7 (23.7; 33.6)	26.4 (22.7; 31.2)	0.002
Waist (cm)	628 (63%)	88 (78; 103)	90 (78; 106)	88 (77; 101)	0.077
FG-score	882 (88%)	11 (5; 15)	12 (7; 16)	10 (5; 15)	0.005
Total testosterone (nmol/L)	690 (69.1%)	1.7 (1.3; 2.4)	1.7 (1.3; 2.3)	1.7 (1.2; 2.4)	0.791
SHBG (nmol/L)	923 (92%)	44 (31; 66)	44 (29; 64)	44 (31; 67)	0.454
Free testosterone (nmol/L)	680 (68.136%)	0.033 (0.021;0.048)	0.034 (0.022;0.048)	0.032 (0.021;0.048)	0.418
Fasting blood glucose (mmol/L)	460 (46.1%)	4.6 (4.2; 5.0)	4.6 (4.3; 5.0)	4.6 (4.2; 5.0)	0.412
	***N* (%)**		***N* (%)**	***N* (%)**	***p*¤**
BMI ≥ 25 kg/m^2^	562 (60.4%)		187 (66.1%)	375 (58.0%)	0.020
BMI < 25 kg/m^2^	368 (39.6%)		96 (33.9%)	272 (42.0%)	0.020
Waist ≥ 88 cm	326 (51.9%)		98 (56.3%)	228 (50.2%)	0.171
Waist < 88 cm	302 (48.1%)		76 (43.7%)	226 (49.8%)	0.171
FG score ≥ 7	628 (71.2%)		206 (77.2%)	422 (68.6%)	0.010
FG score < 7	254 (28.8%)		61 (22.8%)	193 (31.4%)	0.010

# Non-parametric test on the equality of medians. *p*¤ Chi squared test; FG-score: Ferriman Gallwey score.

**Table 6 biomedicines-10-02396-t006:** Crude and adjusted Hazard ratios in PCOS OUH and development of depression (*N* = 930).

		Depression			
	Crude HR (95% CI)	*N*	*p*-Value	Age and BMI Adjusted HR ^a^ (95% CI)	*p*-Value
Age (years)	1.00 (0.99; 1.02)	930	0.590	1.00 (0.99; 1.01)	0.837
BMI (kg/m^2^)	1.02 (1.01; 1.04)	930	0.003	1.02 (1.01; 1.04)	0.003
Waist (cm)	1.01 (1.00; 1.02)	616	0.014	1.03 (1.00; 1.05)	0.022
FG-score	1.02 (1.00; 1.03)	834	0.043	1.02 (1.00; 1.03)	0.041
Total testosterone (nmol/L)	0.97 (0.85; 1.10)	649	0.619	0.95 (0.83; 1.09)	0.467
SHBG (nmol/L)	1.00 (1.00; 1.00)	863	0.525	0.95 (0.83; 1.09)	0.467
Free testosterone (nmol/L)	0.58 (0.04; 8.08)	639	0.685	0.19 (0.01; 3.91)	0.280
Fasting blood glucose (mmol/L)	1.20 (1.01; 1.42)	441	0.037	1.18 (1.00; 1.39)	0.055

Baseline characteristics in PCOS OUH and risk of development of depression. FG-score: Ferriman Gallwey score, SHBG: Sex hormone-binding globulin. Hazard ratios are presented for crude models and models corrected for age and BMI. ^a^ Except age which is adjusted for BMI alone, and BMI which is adjusted for age alone.

## Data Availability

Data is contained within the article.

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
