# Peer review of "Diabetes Mellitus Mediates Risk of Depression in Danish Women with Polycystic Ovary Syndrome—A National Cohort Study"

_biomedicines, 2022, doi:10.3390/biomedicines10102396_

Round 1
Reviewer 1 Report
In the introduction, multiple studies are cited that investigated a link between PCOS and depression. Studying this same question here is repetitive for the existing knowledge, i.e., the study descripted in the manuscript is not particularly innovative. Please high-light how the present study differs from the earlier studies and what new information it provides.
The study investigated development of depression during the follow up that was on average between 7.0 and 11.3 years. Women who were diagnosed with depression prior to the study were excluded. Although on one hand this makes sense, on the other hand it is not clear why PCOS patients would start developing/exhibiting depression only at the mean age of 28. Although the diagnosis of PCOS occurs at that age, there is no reason patients would not have exhibited PCOS-related depression already before they were diagnosed.
Since about the same percentage of control and PCOS patients are nulliparous, but infertility is twice as high in PCOS patients, is it possible that these patients are seeking treatment for infertility much more frequently than the controls because they know PCOS reduces their fertility?
Other comments:
Line 38. Please explain what SF-36 score is
Line 84. Please provide the full name of CPR (presumably in Danish).
Line 221. % is missing after 50.3
Line 319. It is stated that patients with milder depressive symptoms should be treated with psychotherapy. The reference provided to support this statement (39) is entitled “Effects of diet and metformin administration on sex hormone-binding globulin, androgens, and insulin in hirsute and obese women”. According to current recommendations, mild depression should be treated with exercise, diet, social connection, and gratitude. Psychotherapy does not seem to be a standard of care for mild depression.
Author Response
Reviewer 1:
In the introduction, multiple studies are cited that investigated a link between PCOS and depression. Studying this same question here is repetitive for the existing knowledge, i.e., the study descripted in the manuscript is not particularly innovative. Please high-light how the present study differs from the earlier studies and what new information it provides.
We agree and have made wording clearer to emphasize novelty of our study.
The study investigated development of depression during the follow up that was on average between 7.0 and 11.3 years. Women who were diagnosed with depression prior to the study were excluded. Although on one hand this makes sense, on the other hand it is not clear why PCOS patients would start developing/exhibiting depression only at the mean age of 28. Although the diagnosis of PCOS occurs at that age, there is no reason patients would not have exhibited PCOS-related depression already before they were diagnosed.
Yes, we agree. We tested the hypothesis of higher risk of diagnosis of depression after the diagnosis of PCOS. We have added further considerations about study design in the limitations section.
Since about the same percentage of control and PCOS patients are nulliparous, but infertility is twice as high in PCOS patients, is it possible that these patients are seeking treatment for infertility much more frequently than the controls because they know PCOS reduces their fertility?
Yes, this is correct – this is also the reason for correcting results for infertility in regression analyses. We have made this clearer in the Introduction section.
Other comments:
Line 38. Please explain what SF-36 score is
Of course – we have added this.
Line 84. Please provide the full name of CPR (presumably in Danish).
Of course – we have added this.
Line 221. % is missing after 50.3
Sorry, this has been adjusted.
Line 319. It is stated that patients with milder depressive symptoms should be treated with psychotherapy. The reference provided to support this statement (39) is entitled “Effects of diet and metformin administration on sex hormone-binding globulin, androgens, and insulin in hirsute and obese women”. According to current recommendations, mild depression should be treated with exercise, diet, social connection, and gratitude. Psychotherapy does not seem to be a standard of care for mild depression.
Sorry – we have inserted a relevant citation regarding treatment of depression in primary care and have adjusted wording of the sentence.
Reviewer 2 Report
The authors present the risk of depression in PCOS patients using a national database. The data source seems reliable, and the number of cases is enough to show the author's theme. This manuscript will provide good suggestions to clinicians. As the authors mentioned in the discussion, a further study including the cases with pre-diagnosed depression is expected.
There is only one point to improve the manuscript. In Figure 1, the characters in the figure are hard to read. Please make the figures more high resolution.
Author Response
Thank you very much for this nice comment - we have uploaded a new figure to be inserted into the manuscript.
Reviewer 3 Report
1- Add the keywords, I cannot see any keywords.
2- The manuscript lacks the literature citation of some highly interesting most recent relevant works, and thus the reference citations are not up to date. For example Recent progress in polymeric non-invasive insulin delivery. International Journal of Biological Macromolecules. 2022 Jan 29.
3- The manuscript contains numerous grammatical and formatting errors. Please improve the grammar and the language throughout the manuscript.
4- I do not see significant data compared to the result section. Compare all the results with the results of published studies.
5- The authors should prepare Figure 1 with better resolution.
Author Response
Reviewer 2:
1_ Add the keywords, I cannot see any keywords.
Of course – keywords have been added.
2- The manuscript lacks the literature citation of some highly interesting most recent relevant works, and thus the reference citations are not up to date. For example Recent progress in polymeric non-invasive insulin delivery. International Journal of Biological Macromolecules. 2022 Jan 29.
Of course – we have revised and included more recent references.
3- The manuscript contains numerous grammatical and formatting errors. Please improve the grammar and the language throughout the manuscript.
We agree – the manuscript has been converted into the template of the paper without our proofreading. We have adjusted the manuscript to improve readability and grammar.
4- I do not see significant data compared to the result section. Compare all the results with the results of published studies.
We have discussed our results as part of the Discussion section – the Results section only includes our own results. We hope that you agree that this is the best way to present data.
5- The authors should prepare Figure 1 with better resolution.
Yes of course – we have contacted the Biomedicines office and have asked them to insert a figure 1 with improved readability.